# Field Measurements and Analysis of Indoor Environment, Occupant Satisfaction, and Sick Building Syndrome in University Buildings in Hot Summer and Cold Winter Regions in China

**DOI:** 10.3390/ijerph20010554

**Published:** 2022-12-29

**Authors:** Jiantao Weng, Yuhan Zhang, Zefeng Chen, Xiaoyu Ying, Wei Zhu, Yukai Sun

**Affiliations:** 1School of Spatial Planning and Design, Zhejiang University City College, Hangzhou 310015, China; 2Zhejiang Engineering Research Center of Building’s Digital Carbon Neutral Technology, Hangzhou 310015, China; 3School of Economics and Management, China University of Mining and Technology, Xuzhou 221116, China; 4Huahui Engineering Design Group Co., Ltd, Shaoxing 312000, China; 5Hangzhou Integrative Medicine Hospital, Hangzhou 310064, China

**Keywords:** indoor environment quality, occupant satisfaction, sick building syndrome, classroom, green building, retrofitted

## Abstract

Teachers and students work and study in classrooms for long durations. The indoor environment directly affects the health and satisfaction of teachers and students. To explore the performance differences between green buildings, conventional buildings, and retrofitted buildings in terms of their indoor environment, occupant satisfaction, and sick building syndrome (SBS), as well as the correlation between these different aspects, three university teaching buildings were selected in hot summer and cold winter regions in China. These included a green building (GB), a retrofitted building (RB), and a conventional building (CB). Long-term indoor environment monitoring and point-to-point measurements were conducted during the transition season and winter and the indoor environment, satisfaction, and SBS in the three buildings were compared. A sample of 399 point-to-point questionnaires was collected. A subjective-objective indoor environmental quality (IEQ) evaluation model for schools in China was established, covering satisfaction and the indoor environment. The results showed that the compliance rate of the indoor environment in the GB and RB was generally superior to that of the CB. The overall satisfaction was the highest for the GB, followed by the CB, and then the RB. The GB had the highest overall indoor environment quality score, followed by the RB and then the CB. The occurrence of SBS was lowest in the CB, followed by the GB, and then the RB. It was determined that the design of natural ventilation should be improved and that building users should be given the right to autonomous window control and temperature control. To reduce the occurrence of SBS symptoms, attention should be paid to the control of temperature and CO_2_ concentration. To improve learning efficiency, it suggests reducing indoor CO_2_ concentrations and improving desktop illuminance. This study provides a reference for improving the indoor environment and health performance of existing university teaching buildings.

## 1. Introduction

China is one of the largest developing countries in the world and has an unprecedented scale of education. In 2020, the building area of common universities and colleges was 920 million square metres [1]. Many existing teaching buildings in China were built in the 1980s and 1990s; their hardware and software generally cannot meet the high standards of modern users and the indoor environment quality is generally poor [2]. The quality of the indoor environment directly affects the working quality and health of teachers and students, so there is an urgent need for renovation.

The indoor environment of buildings built in different decades varies greatly. Various countries have widely promoted the green building evaluation system to improve building performance, for instance, Leadership in Energy and Environmental Design (LEED), Building Research Establishment Environmental Assessment Method (BREEAM), and Comprehensive Assessment System for Built Environment Efficiency (CASBEE), but the application results are still controversial. One study showed that users of green buildings were more forgiving of their buildings [3]. Another study in the United States suggested that LEED buildings were superior to non-LEED ones in terms of most aspects of the building performance [4]. A post-occupancy evaluation (POE) of 12 green and 12 conventional office buildings across Canada and the northern United States showed that green buildings generally performed better than conventional buildings, including in terms of environmental satisfaction, satisfaction with thermal conditions, and satisfaction with the view to the outside [5]. Long-term and instantaneous measurements as well as point-in-time surveys in four Swiss green buildings showed that temperature and air quality were the most critical factors, and personal control, gender, and climate had an impact on occupant comfort [6]. A study from Jordan [7] indicated that there was no significant improvement in air quality and visual or acoustic satisfaction after occupants moved to a green building. Studies in Singapore revealed that the indoor environment of green buildings was better than that of non-green buildings, and the performance goal of green buildings could be achieved by the green refurbishment of conventional buildings [8]. A review paper concluded that user satisfaction with green buildings in western countries, such as the United States [9], was not significantly different from their satisfaction with traditional buildings, while user satisfaction with green buildings in East Asia, such as China and South Korea, was significantly higher than their satisfaction with traditional buildings.

For teaching buildings, the personnel density is relatively high, and the service time is quite long. The association between the indoor environment and health symptoms, learning efficiency, and satisfaction are the focus of current research [10]. However, as different factors influence each other, the interaction relationships are complex.

Many studies have focused on the impact of the indoor environment on occupant satisfaction and thermal comfort in schools [11,12,13,14]. For instance, a study in the UK [15] highlighted that better indoor air quality could make up for the discomfort caused by higher temperatures, and dissatisfaction with one aspect of indoor environmental quality (IEQ) did not necessarily lead to overall discomfort unless that aspect was grossly unacceptable. Another study determined that the thermal comfort prediction model for educational buildings should consider the influence of gender, age, and socioeconomic background [16]. In addition, mixed-mode ventilation, coupling natural and mechanical ventilation, had the advantages of lower energy and improved thermal comfort [16]. A study of UK primary schools found that the operative and outdoor temperatures during the non-heating season and indoor and outdoor humidity during the heating season were the main predictors of the open window area, while the main driver for the operation of external doors was occupancy patterns [17].

Learning performance is one of the most important concerns in school IEQ studies. A comparison study suggested that occupants’ cognitive scores were higher on the green building day than on the conventional building day and that VOCs and CO_2_ were independently associated with cognitive scores [18]. A study under summer conditions in China showed that when participants felt ‘slightly warm’, their learning performance was highest [19]. Real-scale experiments in a meeting room revealed that the overall performance declined by 1% for every 10 μg/m^3^ increase in PM_2.5_ concentration [20]. A series of measurements in 220 classrooms in the USA revealed that the type of mechanical system and an adequate ventilation rate played an important role in classroom indoor air quality and were significantly related to student learning outcomes [21]. In addition, proper ventilation and effective filters help dilute contaminants and potentially improve student performance.

Recently, an increasing number of studies have been conducted to analyse the impact of the indoor environment on health symptoms. An increased risk of allergy and flu-like symptoms was associated with hot classrooms during the heating season, an increased risk of asthma-like symptoms was associated with noisy classrooms [22], and a protective effect for allergy was associated with good outdoor air quality. A comparison study showed that participants in green buildings experienced improved IEQ compared to those in conventional buildings and, as a result, they also reported fewer symptoms [23]. However, another study tracked the transition from conventional office buildings to green buildings and found that the percentage of occupants who experienced fever and flu symptoms increased by 10% [7]. SBS symptoms vary depending on comfort conditions such as hygiene, ventilation, and heating instead of the age of the school building. School principals responsible for the administration of school buildings have a marked impact on the improvement or deterioration of SBS symptoms [24]. Results in Iran showed that psychological factors such as job satisfaction, working environment, working hours, and communication with colleagues/employers were the most important factors affecting the prevalence of sick building syndrome [25]. Another study in northern Iran suggested that there were significant correlations between CO_2_ and temperature with SBS symptoms [26]. A study from the eastern Mediterranean climate in educational laboratories indicated that SBS symptoms were associated significantly with education year and gender [27]. A higher CO_2_ concentration was significantly associated with a higher percentage of perceived stuffy odour and skin SBS symptoms in Chinese homes [28]. A study in Japan depicted that allergies and lifestyle behaviors were associated with increased SBS in children, including skipping breakfast, displaying faddiness, constipation, insufficient sleep, not feeling refreshed after sleep, and the lack of deep sleep [29].

In the post-occupancy evaluation of school buildings, previous studies mostly adopted retrospective investigations or climate chamber comparative experiments, while comparison studies of green and non-green buildings mainly focused on offices [9]. Few studies have used the point-to-point field measurement method to characterise the relationship between the indoor environment and other performances [18]. Results that are obtained through objective and subjective approaches could differ, so it is necessary to combine these approaches to comprehensively assess the IEQ [30]. During point-to-point measurements, the indoor environment, SBS, and various occupant satisfaction measures were monitored simultaneously during the operation stage while carrying out the questionnaire survey [31]. There has been a lack of mutual verification of subjective and objective data in the performance comparison of different kinds of buildings, such as occupant satisfaction, and the influencing mechanism between these data has not been accurately analysed. Additionally, there have been few comparative studies on different kinds of university teaching buildings in hot summer and cold winter regions in China.

The goals of this study were:

(1) To analyse and compare the indoor environment, occupant satisfaction, self-reported SBS, and learning efficiency of green, retrofitted, and conventional college classrooms in hot summer and cold winter regions of China more accurately, through a combination of point-to-point testing and long-term monitoring,

(2) To quantify the impact of the indoor environment on student satisfaction and SBS symptoms, extract the key indoor parameters and personal factors that affected occupant satisfaction, SBS symptoms, and learning efficiency, and provide advice for the design of retrofitting university classrooms.

## 2. Methods

### 2.1. Case Study Context

Most of the existing university buildings in China were built in the 20th century. In 2006, the first version of the China Evaluation Standard for Green Buildings (CESGB) was promulgated, and most green university buildings that met the requirement of the CESGB, LEED, or BREEAM in China were built after 2010. To reflect the representativeness of all the building samples, three teaching buildings of a similar size in hot summer and cold winter areas in China were selected to carry out field measurements, as shown in Table 1. The green building (GB) was completed in 2017 and was verified by LEED V4.1 O+M (Green Building Council, Washington, DC, USA). The conventional building (CB) was completed in 1999. The retrofitted building (RB) was completed in 2010 and the interior and exterior renovation was completed in 2012.

The layout of a standard floor and measurement points of the three teaching buildings are shown in Figure 1. The GB was in the suburb of Jiaxing, Zhejiang and the classrooms were arranged around the atrium. Its daily average number of users was about 100. Both the RB and CB teaching buildings were in the city centre of Hangzhou, Zhejiang. The classrooms in the RB were arranged from north to south, with corridors in the middle, and the daily average number of users was about 300. In the CB, the classrooms were arranged in the north and corridors in the south, and the daily average number of users of the building was about 150. GB is about 60 km away from RB and CB. RB and CB are on the same campus. The changes in outdoor air temperature in Hangzhou are shown in Figure 2. The average temperature in January is 7.4 ℃, the lowest is 0 ℃, the average temperature in July is 27.2 ℃, and the highest is 37.0 ℃; July is sunny and hot with little rain and often dry. Annual precipitation is about 1500 mm. These three buildings reflect the evolution of college and university teaching buildings in hot summer and cold winter regions in China.

### 2.2. Data Collection

#### 2.2.1. Indoor Environmental Measurements

This study was approved by the ethics committee of Hangzhou Integrative Medicine Hospital. As the classrooms were not used in the summer, indoor environmental measurements were taken during the transition season and winter. The transition season test time was from October to November 2020, and the winter test time was from December 2020 to January 2021. The IEQ generally includes measures of the thermal, light, air quality, and acoustic environment [11,12,13,14]. Considering the required parameters and instrument availability, the indoor environment test parameters in this study included air temperature, relative humidity, PM_2.5_ concentration, CO_2_ concentration, desktop illuminance, and sound pressure level.

The measurement plan of the indoor environment is shown in Table 2. A self-developed machine was used, which was placed in the centre of the classroom to record the air temperature, relative humidity, CO_2_ concentration, PM_2.5_ concentration, and sound pressure level every 5 minutes. The single-chip computer NUC131LD2AE was used as the control core of the whole system, as shown in Figure 3. The sensors used in the machine and their accuracies are shown in Table 3. Considering the different orientations and room sizes, four classrooms were selected in the GB, four in the RB, and two in the CB. Since desktop illuminance was greatly affected by human activities and natural lighting, this measurement was obtained by the investigators during the point-to-point test, when artificial lighting was used.

To quantify the performance of the indoor environment, the compliance rate of the indoor environment was taken as the basic quantitative index. Two weeks of indoor environmental data in typical seasons were selected to represent the indoor environment performance in the corresponding seasons. The calculation of the compliance rate only considered the indoor environment data during working hours, which were defined as Monday to Friday, 8:00–21:00. The typical weekly compliance rate of indoor environmental parameters was calculated according to Equations (1) and (2):(1)$$P_{j}=100\%\times \sum_{i=1}^{N_{j}}S_{i,j}/N_{j}$$
(2)$$S_{i,j}=\left\{\begin{array}{l} 0,\;\mathrm{data}\;\mathrm{fail}\;\mathrm{to}\;\mathrm{meet}\;\mathrm{the}\;\mathrm{requirement}\;\mathrm{of}\;\mathrm{standard}\\ 1,\;\mathrm{data}\;\mathrm{meet}\;\mathrm{the}\;\mathrm{requirement}\;\mathrm{of}\;\mathrm{standard} \end{array}\right.$$

Here, *N_j_* is the total amount of calculated indoor environment parameter *j*, *P_j_* is the compliance rate of indoor environment parameter *j*, and *S_i_* is the calculated result of the measuring point data. The result was based on relevant national standard requirements (Table 4).

The variation coefficient was introduced to quantify the degree of fluctuation of the indoor environmental data. The calculation method is shown in Equation (3).
(3)α=SD/X

Here, *X* is the average value of the environmental parameters, *SD* is the standard deviation of the environmental parameters, and *α* is the coefficient of variation of the environmental parameters.
ijerph-20-00554-t004_Table 4Table 4Calculation basis of compliance rate of the indoor environment.ParameterTransitionWinterThe Reference StandardAir temperature21–28 °C18–24 °CGB 50736-2012 [32]Relative humidity-≥30%GB 50736-2012 [32]CO_2_ concentration≤1000 ppmGB/T 18883-2022 [33]PM_2.5_ concentration≤37.5 µg·m^−^^3^T/ASC 02-2021 [34]Illuminance≥300 luxGB 50034-2013 [35]Acoustic≤45 dB (*L*_Aeq,T_)GB 55016-2021 [36]


#### 2.2.2. Questionnaire Survey

Point-to-point measurements and surveys were conducted in the three buildings simultaneously. That is, the satisfaction and SBS questionnaire survey was carried out while indoor environmental parameters were recorded. The content of the questionnaire included occupant background information, occupant satisfaction, and self-reported SBS symptoms. The occupant environmental satisfaction included nine aspects: the temperature, humidity, air quality, visual environment, acoustic environment, overall environment, learning efficiency, interior space design and building service performance. Self-reported SBS symptoms included 11 common symptoms, as shown in Table 5.

To ensure the reliability and accuracy of the results, more than 30 questionnaires were required in each building throughout the year. A 7-level Likert scale [37] for satisfaction was adopted in this study according to the CBE Survey [38]. The scale from 1 to 7 represents “very dissatisfied”, “dissatisfied”, “somewhat dissatisfied”, “neutral”, “relatively satisfied”, “satisfied”, and “very satisfied”. For SBS symptoms, participants were asked to respond yes or no.

#### 2.2.3. IEQ Evaluation Method

A comprehensive evaluation method for the indoor environment was proposed considering the four aspects of thermal, air quality, and the visual and acoustic environment, as shown in Equations (4)–(8).
(4)$$IEQ_{T}=100\times \bar{P_{T}}$$
(5)$$IEQ_{\mathrm{AQ}}=\frac{100\times \left(\bar{P_{\mathrm{CO}2}}+\bar{P_{\mathrm{PM}2.5}}\right)}{2}$$
(6)$$IEQ_{V}=100\times \bar{P_{V}}$$
(7)$$IEQ_{\mathrm{AC}}=100\times \bar{P_{\mathrm{AC}}}$$
(8)$$IEQ_{\mathrm{overall}}=IEQ_{T}w_{T}+IEQ_{\mathrm{AQ}}w_{\mathrm{AQ}}+IEQ_{V}w_{V}+IEQ_{\mathrm{AC}}w_{\mathrm{AC}}$$

Here, *IEQ*_T_ is the thermal environment score, *IEQ*_AQ_ is the air quality score, *IEQ*_V_ is the visual environment score, *IEQ*_AC_ is the acoustic environment score, *IEQ*_overall_ is the overall environment score, *w*_T_ is the thermal environment weight coefficient, *w*_AQ_ is the air quality weight coefficient, *w*_V_ is the visual environment weight coefficient, and *w*_AC_ is the acoustic environment weight coefficient.

The multiple linear regression model was used to describe the relationship between the indoor environment factor and overall IEQ [39] and to determine the weight coefficient of each indoor environment factor. The multiple linear regression model is shown in Equation (9).
(9)$$y_{i}=\bar{y_{i}}+e_{i}=b_{0}+b_{1}x_{1}+\cdots +b_{n}x_{n}+e_{i}$$

Here, *y_i_* is the measured value and $\bar{y_{i}}\;$is the predicted value, when the respective variables were determined. The predicted value represents the part that can be determined by the independent variable; *x*_1_, *x*_2_…, *x_n_* are the independent variables; *e_i_* is the residual, which is the difference between the measured value of the dependent variable *y_i_* and its predicted value $\bar{y_{i}}$, representing the part not determined by the independent variable; *b*_0_ is a constant, which represents the estimated value of the dependent variable when all the independent variables are 0.

### 2.3. Statistical Analysis

Since the satisfaction questionnaire data could not meet the normal distribution requirements of the t-test, the Mann–Whitney U test was used to compare the differences in satisfaction among the different buildings in different seasons. The Pearson correlation analysis was used to determine the correlation between indoor environmental parameters, occupant satisfaction, and the occurrence of SBS. The Mann–Whitney U test was used to determine the impact of personal background information, seat location, and other factors on the occurrence of SBS. All statistical analyses were performed using SPSS version 20 (IBM Inc., Armonk, NY, USA).

## 3. Results

### 3.1. Long-Term Indoor Environment

The results of the indoor environment measurements are shown in Figure 4, including the distribution and variation coefficients of parameters relating to the indoor thermal environment, air quality visual environment and acoustic environment in the GB, RB, and CB.

#### 3.1.1. Thermal Environment

Air Temperature

In general, the indoor temperature fluctuation gap of the three buildings in the transition season was small and the average indoor temperature of the GB was 2 °C higher than that of the RB and CB. The temperature of the GB was controlled at about 22.5 °C with little temperature fluctuation. The highest temperature in the GB, RB, and CB was about 23 °C, 22 °C, and 22.5 °C, respectively.

In winter, the variation coefficient of temperature for the CB was significantly larger than that of the RB and GB. The minimum temperature in the CB was 6 °C, which was much lower than that of the GB and RB (16 °C). This was because the CB was in a state of natural ventilation for long durations in winter and its exterior wall was not equipped with an insulation layer.
Relative Humidity

During the transition season, the relative humidity in the three buildings ranged from 35% to 75%. The relative humidity of the GB was maintained within the standard range, while the average relative humidity of the RB and CB was about 5% higher than that of the GB. The variation coefficients of the RB and CB were about 0.1 higher than that of the GB.

In winter, the indoor relative humidity of the GB was within a reasonable range and the relative humidity of the RB and CB could not meet the standard requirements for half of the time. The variation coefficient of relative humidity for the RB was significantly greater than that of the GB. The relative humidity of the RB ranged from 15% to 68%, which was very unstable.

Mechanical ventilation was used in the GB throughout the year and the windows were closed for long durations, so the indoor air humidity in winter and the transition season was relatively stable. However, as mixed ventilation was used in the RB and natural ventilation in the CB, the indoor humidity of these buildings fluctuated greatly.

#### 3.1.2. Indoor Air Quality

CO_2_ Concentration

In general, the CO_2_ concentration in the three buildings was below the standard limit of 1000 ppm. In transition season, on average, the CO_2_ concentration of the GB was about 140 ppm higher than that of the RB and 190 ppm higher than that of the CB. The peak concentration of CO_2_ in the GB reached 1500 ppm, which was significantly higher than that in the RB and CB. The variation coefficient of CO_2_ in the GB was almost three times higher than that of the RB and GB. In winter, the indoor CO_2_ concentration of the GB, RB, and CB decreased successively, and the CO_2_ concentration of the CB changed by around 400 ppm, which was near the outdoor CO_2_ concentration. The variation coefficient of indoor CO_2_ concentration of the RB reached 0.75, which was significantly higher than that of the CB. This was possibly because the windows in the CB were open for long durations; that is, the building was in a state of natural ventilation.
PM_2.5_ Concentration

The PM_2.5_ concentration of the GB was lowest in the transition season and winter, with average values of 32.5 and 30 μg/m^3^, respectively. The average PM_2.5_ concentrations of the RB and CB exceeded the national guideline of 37.5 μg/m^3^. In the GB, a fresh air system was employed, and the windows were closed most of the time. Furthermore, the GB was in a suburb of the city, so the outdoor PM_2.5_ level was relatively low. A mixed ventilation mode was used in the RB building. Despite the employment of the fresh air system, it was still susceptible to the outdoor PM_2.5_ concentrations when the windows were open, while the CB was in a state of natural ventilation for long durations and its PM_2.5_ concentration was highly correlated with the outdoor PM_2.5_ concentration in winter. The indoor/outdoor PM_2.5_ comparisons in Figure 5 showed that the indoor PM_2.5_ of GB was lower than the outdoor level for a long time, while the indoor/outdoor PM_2.5_ values in RB and CB fluctuated around one. GB of indoor PM_2.5_ was less affected by outdoor, possibly because its windows had better air tightness.

#### 3.1.3. Visual Environment

The desktop illuminance test was carried out at night, in the absence of natural lighting. The desktop illuminance of the GB was better than that of the RB and CB. The average desktop illuminance of the GB reached 450 lux, while that of the RB and CB was less than 300 lux. There was little difference between the illuminance in winter and that in the transition season. The difference in illuminance was mainly due to the lighting design and lamp selection.

#### 3.1.4. Acoustic Environment

The acoustic environment of the GB was significantly better than that of the RB and CB. In the GB, the A-weighted sound pressure level generally met the requirements of the standard. However, the acoustic environment of the RB and CB were influenced by outdoor traffic noise and human activities, and their variation coefficients of noise were significantly greater than that of GB. This was because the RB and CB were in the city centre, and RB was close to the main road.

#### 3.1.5. Compliance Rate of the Indoor Environment

The compliance rate of six indoor environmental parameters was obtained, including air temperature, relative humidity, A-weighted sound pressure level, desktop illuminance, CO_2_ concentration, and PM_2.5_ concentration, as shown in Table 6.

In the transition season, the GB performed better than the RB and CB (except for the CO_2_ concentration). Its air temperature compliance rate was 98.7%, while that of the RB and CB was only 39.8% and 47.8%, respectively. The relative humidity compliance rate of all three buildings was above 80%. The CO_2_ concentration compliance rate in the GB was 78.5%, while that of the RB and CB was 100%. In terms of the PM_2.5_ concentration, the GB performed better than the RB and CB. The desktop illuminance compliance rate in the GB was 100%, while that of the RB and CB was only 40.4% and 21.6%, respectively. The acoustic environment compliance rate of the GB was significantly higher than that of the RB and CB.

In winter, the air temperature compliance rate of the RB was higher than that of the GB and CB. In terms of air temperature, the GB achieved 20.6%, while RB and CB achieved 79.8% and 36.7%, respectively. The CB achieved the highest compliance rate for CO_2_ concentration at 98.1%, compared to the GB (85.4%) and RB (82.3%). The indoor PM_2.5_ concentration compliance rate of the CB was only 38.0%, which was significantly lower than that of the GB and RB. The illuminance compliance rate of the RB and CB was 0%. The sound pressure level compliance rate of the GB was 79.6%, while that of the RB and CB was only 58.7% and 46.4%, respectively.

### 3.2. Occupant Satisfaction

In the transition season and winter, 30 and 35 valid questionnaires were collected in the GB, 100 and 50 valid questionnaires were collected in the RB, and 134 and 50 valid questionnaires were collected in the CB, respectively. Background information on all the respondents is summarised in Table 7. The respondents were all college students studying in the classrooms, 59% of whom were females, and most were between 18 and 22 years old.

Figure 6 summarises the occupant satisfaction results for the three buildings in the transition season and winter, including relating to the thermal environment, humidity, visual and acoustic environment, air quality, and service performance. The significant levels of satisfaction obtained by the Mann–Whitney U test are also presented.

#### 3.2.1. Thermal Satisfaction

Temperature Satisfaction

In the transition season, the highest temperature satisfaction was with the GB, followed by the CB and RB. The results of the Mann–Whitney test showed that the temperature satisfaction with the GB was significantly better than that with the RB in the transition season and the temperature satisfaction with the CB was higher than with the RB. In winter, the average temperature satisfaction was for the GB (5.5), followed by the CB (5.2) and RB (5.1). The differences between the three buildings were smaller than in the transition season.
Humidity Satisfaction

Figure 6b compares occupant satisfaction with the GB, RB, and CB in the transition season and winter. The highest humidity satisfaction was with the GB in the transition season, and the humidity satisfaction with the CB was higher than that with the RB. There were significant differences in the humidity satisfaction with the three buildings. The humidity satisfaction in winter from highest to lowest was GB (5.3) > CB (4.6) > RB (4.3). The gap between each value was smaller than in the transition season and there was a significant difference between the GB and RB.

#### 3.2.2. Visual Satisfaction

Figure 6c compares occupant visual satisfaction with the GB, RB, and CB in the transition season and winter. In the transition season, there was greater visual satisfaction with the GB than with the RB and CB, for which the average visual satisfaction values were 5.5, 5.1, and 5.1, respectively. There was a significant difference between the GB and RB, CB. The visual satisfaction with the GB was also higher than with the CB and RB in winter. The satisfaction gap between the GB and RB was slightly larger and there was a significant difference between the GB and RB. In general, the visual satisfaction results were similar in the two seasons.

#### 3.2.3. Acoustic Satisfaction

Figure 6d shows the acoustic satisfaction with the GB, RB, and CB. The average acoustic satisfaction with the GB was higher than with the CB and RB. The average acoustic satisfaction from greatest to least was GB (5.3), CB (5.1), and RB (4.7). There were significant differences between the RB and the other two buildings.

#### 3.2.4. Air Quality Satisfaction

Figure 6e shows the air quality satisfaction with the GB, RB, and CB in the transition season and winter. In the transition season, the average air quality satisfaction from greatest to least was GB (5.4), CB (5.1), and RB (4.5). There were significant differences between RB and the other two buildings. In winter, the air quality satisfaction results for the three buildings were close to those in the transition season but the difference between the RB and CB was smaller.

#### 3.2.5. Overall Environment Satisfaction

Figure 6f compares the overall satisfaction with the GB, RB, and CB during the transition season and winter. In both seasons, the GB showed obvious advantages compared with RB and CB, while the difference between the RB and CB was not significant.

#### 3.2.6. Satisfaction with Building Services

Figure 6g shows satisfaction with learning efficiency, interior space design, and building services in the GB, RB, and CB. In general, the average satisfaction with building services was GB > RB > CB. The satisfaction gaps for learning efficiency and space size were consistent. The GB performed better than the RB and CB, while there was no significant difference between the RB and CB. In terms of operation and maintenance, the GB performed best, followed by the RB and then the CB.

### 3.3. SBS Symptoms

Figure 7 presents the results of the occurrence of SBS symptoms in the buildings throughout the year. Overall, the occurrence of SBS in the GB, RB, and CB was 26.2%, 29.3%, and 24.5%, respectively. The occurrence of “weakness, lethargy”, and “distracted or error-prone” was high, reaching 10% and 12% in the GB, respectively. The occurrence of “distracted or error-prone” in the RB was 12%. The occurrence of “dry, itchy or irritated eyes” in the GB was 4%, in the RB was 11%, and in the CB was 6.5%. The occurrence of “dizziness or headache” in the GB was high, reaching 9%. In the RB, the occurrence of “dry, itchy, runny or bleeding nose” was up to 12%.

### 3.4. The Relationship between Different Aspects

#### 3.4.1. Relationship between the Indoor Environment and the Occurrence of SBS Symptoms

Pearson correlation analysis was used to analyse the relationship between indoor environmental parameters and the occurrence of SBS symptoms (Table 8). The results showed that air temperature, relative humidity, CO_2_ concentration, and PM_2.5_ concentration were directly correlated with the occurrence of SBS symptoms. There was a positive correlation between air temperature and the occurrence of “dry, itchy, runny or bleeding nose” and “sore throat or dry tongue”. The temperature was negatively correlated with the occurrence of “asthma, dry cough, tracheitis” and “flushed, dry or itchy skin”. There was a negative correlation between the relative humidity and the occurrence of “dry, itchy, runny or bleeding nose”. CO_2_ concentration was negatively correlated with the occurrence of “asthma, dry cough, tracheitis” and “flushed, dry or itchy skin”. There was a negative correlation between the PM_2.5_ concentration and the occurrence of “sore throat or dry tongue”.

#### 3.4.2. Relationship between the Indoor Environment and Learning Efficiency

Pearson correlation analysis was used to analyse the relationship between indoor environmental parameters and learning efficiency (Table 9). The results showed that there was a negative correlation between CO_2_ concentration and learning efficiency, while there was a positive correlation between desktop illuminance and learning efficiency. There was no significant correlation between other indoor environmental parameters and learning efficiencies, such as air temperature, relative humidity, PM_2.5_ concentration, and sound pressure level.

#### 3.4.3. Relationship between the Occurrence of SBS Symptoms and Learning Efficiency

Pearson correlation analysis was used to analyse the relationship between SBS symptoms and learning efficiency (Table 10). The results showed that more than half of the SBS symptoms were significantly related to learning efficiency. The symptoms of “dry, itchy, runny or bleeding nose” and “weakness, lethargy” were significantly negatively correlated with learning efficiency.

#### 3.4.4. Relationship between Gender, Seat Location, and the Occurrence of SBS Symptoms

The Mann–Whitney U test was used to analyse the relationship between gender, seat location, and SBS symptoms. The results showed that gender had no significant effect on most symptoms except for “dry, itchy, runny or bleeding nose”, and “dizziness or headache” symptoms. The former was more common in women and the latter was more common in men. It is found that the average occurrence of SBS in students near doors and windows was generally lower than that in students in the middle of the classroom, and the difference in seat location had a significant impact on the occurrence of SBS symptoms.

### 3.5. Comparison of Building Performance in Three Buildings

Based on the point-to-point satisfaction questionnaires in the three buildings, a regression model between overall satisfaction and sub-satisfaction was obtained by multiple regression analysis (Table 11). The variance inflation factor (VIF) of the collinearity representative index was less than 5, so there was no multicollinearity among the variables. In the regression model, the humidity satisfaction did not pass the significance test (*p* < 0.05) but after removing this factor, the regression model of overall building satisfaction and each remaining satisfaction factor was re-established as shown in Equation (10):(10)$$S_{\mathrm{overall}}=1.126+0.247S_{T}+0.103S_{\mathrm{AQ}}+0.134S_{V}+0.265S_{\mathrm{AC}}\;R^{2}=0.483$$

Here, *S*_overall_ is overall satisfaction, *S*_T_ is temperature satisfaction, *S*_AQ_ is air quality satisfaction, *S*_V_ is visual satisfaction, and *S*_AC_ is acoustic satisfaction. The R^2^ value of the regression model was 0.483, indicating that the regression model had a good fitting effect.

The standardised coefficients in Table 11 were normalised and adjusted before comparison and the weighting results of the five factors are shown in Table 12. Among these, humidity failed to pass the significance test and was excluded. Therefore, the IEQ evaluation model included four factors: thermal environment, air quality, light environment, and sound environment.

Based on the weightings shown in Table 12, the *IEQ*_overall_ was calculated. Table 13 compares the compliance rates of the indoor environment, satisfaction, IEQ, and SBS results of the three buildings. In general, the GB performed better in terms of the overall evaluation, IEQ, satisfaction, and occurrence of SBS symptoms than the RB and CB, which was consistent with the findings of previous studies [4,40]. The *IEQ*_overall_ of the GB was the highest at 75.4, followed by the RB (53.5), and then the CB (40.3). In terms of the subjective evaluation, the overall satisfaction with the GB was superior to that with the RB. Notably, although the environmental compliance rate of the RB was higher than that of the CB and its IEQ evaluation result was slightly higher than for the CB, the overall satisfaction with the RB was 0.3 lower than with the CB. This confirmed that subjective and objective assessments can be quite different, which was consistent with the findings of a previous study [30]. This was possibly due to the effect of the pattern of doors and windows [17].

## 4. Discussion

### 4.1. Comparison of IEQ Weights in Different Types of Buildings

Previous studies have shown that the IEQ weights can be influenced by factors such as climate, building type, and sample size [41,42]. In the present study, there were large differences in the IEQ weightings, and the reasons underpinning this were difficult to determine [41,42]. Figure 8 compares the weights of the four IEQ aspects in nine studies. The weighting results for the buildings in this study were close to the weighting results proposed by Cao et al. (2012) for library and teaching buildings in Beijing and Shanghai in China [43]. However, the weightings for the thermal environment in this study were smaller than those proposed by Tahsildoost and Zomorodian [39] for university teaching buildings in Tehran, Iran. The weightings for the acoustic environment were similar to each other. Wei et al. [41] took eight green building evaluation standards as examples and found that the air quality weighting was the largest, followed by the thermal and visual environment weightings, and then the acoustic environment weighting. Additionally, some studies [44,45] subjectively assigned the same weighting to the four indoor environments aspects; that is, 0.25. The average value of the four aspects was taken as the overall IEQ assessment value. 

In conclusion, there were considerable differences in the weightings of the sub-environment in the different climates, and the results were mainly affected by the buildings and test methods. The regression analysis results of the large-scale user questionnaire were inconsistent with the scores of the current GB evaluation standards and the evaluation results of a few previous studies. The results of large-scale user questionnaire analysis generally showed that the thermal environment was the dominant aspect and air quality had the least impact on the IEQ among the four aspects.
Figure 8Comparison of the IEQ weightings from different studies [39,40,41,43,44,46,47,48].
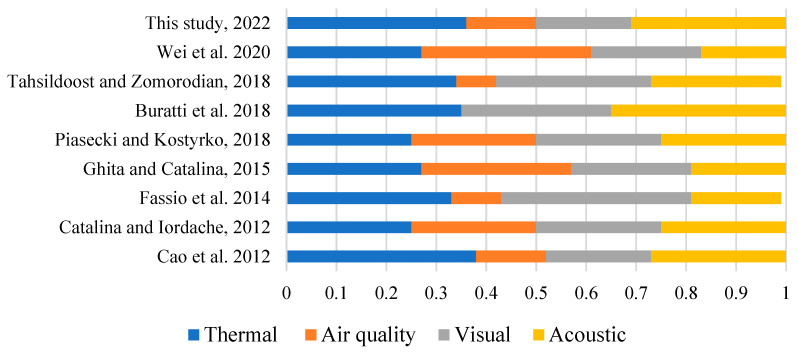



### 4.2. Recommendations for University Teaching Building Design

(1) The renovation of buildings not only needs to improve the hardware conditions but also needs to improve the IEQ in terms of various aspects such as natural ventilation, indoor environment regulation, and property management. As shown in a study from the UK, the operation of windows and external doors directly influences indoor thermal comfort [17] and the PM_2.5_ level [49]. In this study, it was found that the compliance rate of the RB was improved to a certain extent compared with the CB but the improvement effect was not pronounced and the corresponding user satisfaction was even lower than that with the CB. 

(2) The interference of external noise in teaching buildings should be avoided. Noise has a direct impact on the satisfaction of the users in the teaching building, so this is a basic requirement. Users are less tolerant of high levels of noise than other aspects of the indoor environment. The study from Chongqing, China [50] also suggested that acoustic satisfaction was the lowest among the four environmental aspects. Therefore, there should be a focus on ensuring a quiet environment in the classroom and maintaining a safe distance from external noise sources in the planning and design stage.

(3) To reduce the occurrence of SBS, attention should be paid to the control of temperature and CO_2_ concentration. The occurrence of SBS was directly related to air temperature, relative humidity, CO_2,_ and PM_2.5_ concentrations in this study, while in Iran, researchers found that there were significant correlations between CO_2_ and temperature with SBS symptoms [26]. Results from Europe [51] also showed that there was an association between temperature and upper airway symptoms. Another study indicated that gender differences in terms of thermal comfort and SBS symptoms were significant in cool environments but negligible in moderate environments [52]. In addition, another study in a hospital [53] found that gender appeared to be associated with SBS. However, in the present study, few symptoms were correlated with gender and different SBS symptoms had different relationships with temperature and should therefore be considered separately; this finding was consistent with that of a previous study [52]. 

(4) To improve learning efficiency, the indoor CO_2_ concentration should be controlled and desk illuminance improved. The CO_2_ concentration result was consistent with the results of previous studies [20,54]. Regarding the visual environment, a comparison study in healthcare settings confirmed that illuminance was positively related to productivity [4]. High illuminance significantly improved learning performance [55]. Many old teaching buildings in China are unable to meet the requirements of efficient learning due to outdated lighting equipment. Therefore, to achieve good learning efficiency, it is suggested that CO_2_ concentrations be maintained at low levels in the classroom and the desktop illuminance improved. 

### 4.3. Limitations and Future Studies

Only three buildings were measured in this study, so the research results have limitations. Additionally, the operation of university teaching buildings has intermittent characteristics and the use of lighting and air conditioning is directly related to whether students are in the room. Although classrooms followed fixed class schedules, it was difficult to obtain accurate in-room time data during the monitoring process. Accordingly, there was time ambiguity in the calculation of the IEQ compliance rate, so the IEQ may have been underestimated. In the future, it is necessary to obtain simultaneous occupancy data to more accurately describe the relationship between the indoor environment and human behaviour, as well as the actual indoor environment measurements.

Preliminary results indicated that seat location may influence the occurrence of SBS, which may be due to ventilation efficiency. More detailed research is needed on this. In the future, it is necessary to carry out larger-scale research, including in schools with higher personnel density, to determine the impact of different seat locations on the occurrence of SBS symptoms.

## 5. Conclusions

This study conducted a one-year post-occupancy evaluation study on three university teaching buildings to compare the performance differences in terms of the indoor environment, occupant satisfaction, SBS, and other aspects of green, traditional, and renovated university buildings. The relationships between different aspects were analysed. A total of 399 questionnaires and their corresponding environmental parameters were obtained through a point-to-point survey.

The long-term indoor environment compliance rate in the green building was better than that of the renovated and traditional buildings. The compliance rates for the acoustic environment and light environment in the traditional and renovated buildings were quite low. Overall, the green building had the highest overall IEQ, followed by the retrofitted building and then the traditional building.

The green building had the best occupant satisfaction and IEQ, but the traditional building had the best health performance. The retrofitted building did not reveal many advantages compared with the traditional building in terms of occupant satisfaction. The results of the occupant questionnaire analysis generally showed that the thermal environment was the dominant factor and the impact of air quality on the IEQ was the lowest.

There was a negative correlation between CO_2_ concentration and learning efficiency, while there was a positive correlation between desk illuminance and learning efficiency. Gender and seat information had little effect on the occurrence of most SBS symptoms. More than half of the SBS symptoms had a significant relationship with learning efficiency. The relationships between temperature, relative humidity, CO_2_, PM_2.5,_ and SBS symptoms were significant.

In conclusion, the retrofitting process should not only pay attention to the improvement of hardware performance but also the improvement of soft aspects such as ventilation, property management, operations, and maintenance. Teaching buildings should avoid the interference of external noise as far as possible. To reduce the occurrence of SBS, attention should be paid to the control of temperature and CO_2_ concentrations. To improve learning efficiency, attention should be paid to controlling indoor CO_2_ concentrations and improving desk illuminance.

## Figures and Tables

**Figure 1 ijerph-20-00554-f001:**
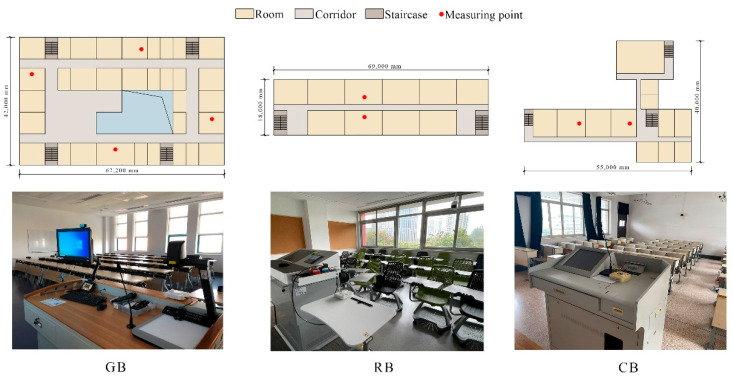
Typical layout of selected buildings and photos of measuring points.

**Figure 2 ijerph-20-00554-f002:**
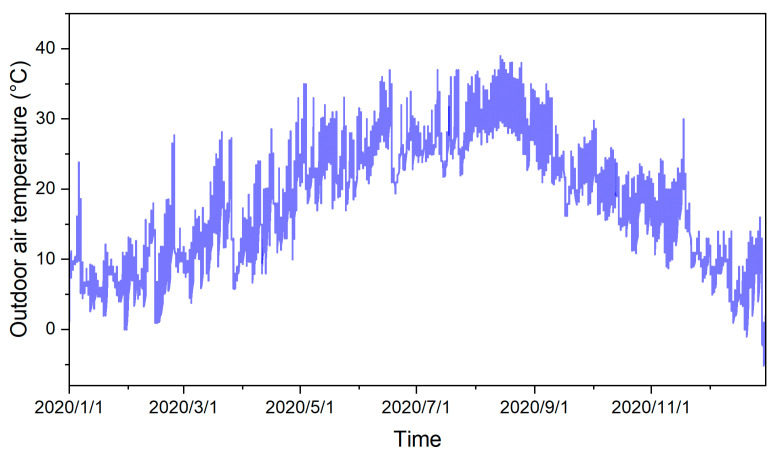
Changes in outdoor air temperature in Hangzhou in 2020.

**Figure 3 ijerph-20-00554-f003:**
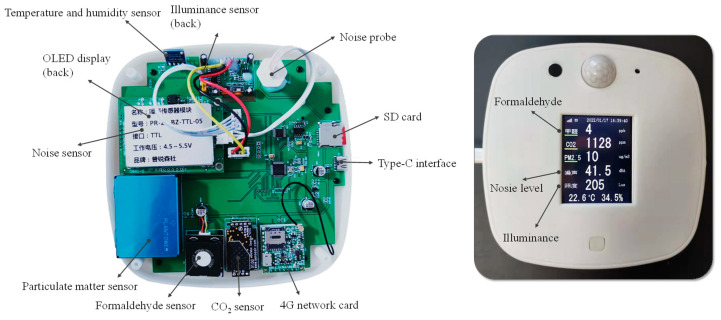
Physical design of the IEQ monitoring instrument.

**Figure 4 ijerph-20-00554-f004:**
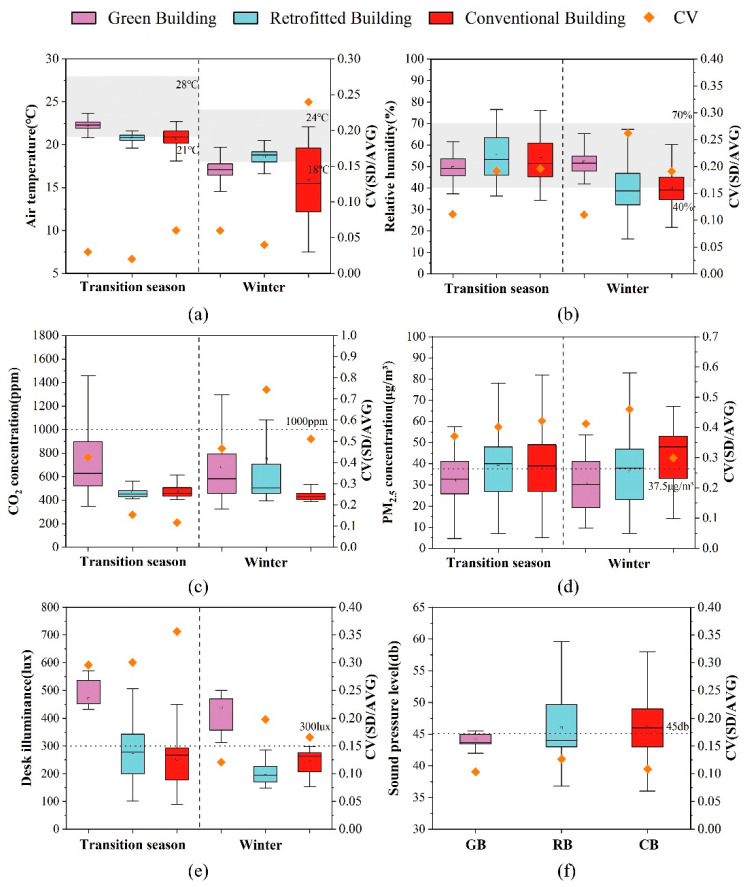
Comparison of (**a**) air temperature, (**b**) relative humidity, (**c**) CO_2_ concentration, (**d**) PM_2.5_ concentration, (**e**) desk illuminance, and (**f**) A-weighted sound pressure level in the GB, RB, and CB.

**Figure 5 ijerph-20-00554-f005:**
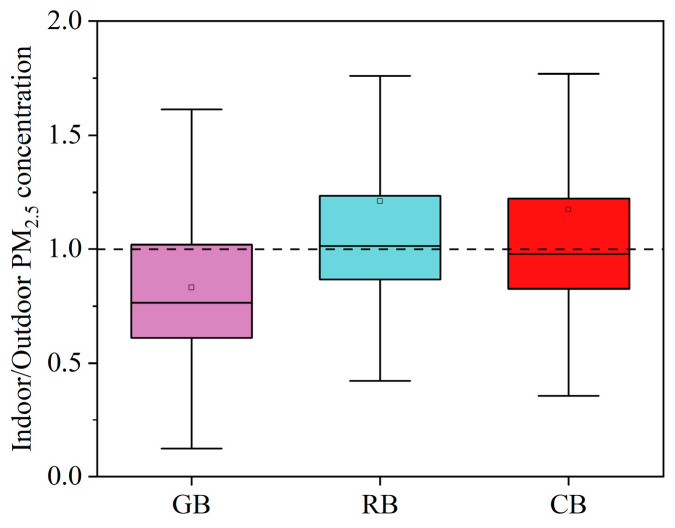
Comparison of indoor/outdoor PM_2.5_ concentrations in the GB, RB, and CB.

**Figure 6 ijerph-20-00554-f006:**
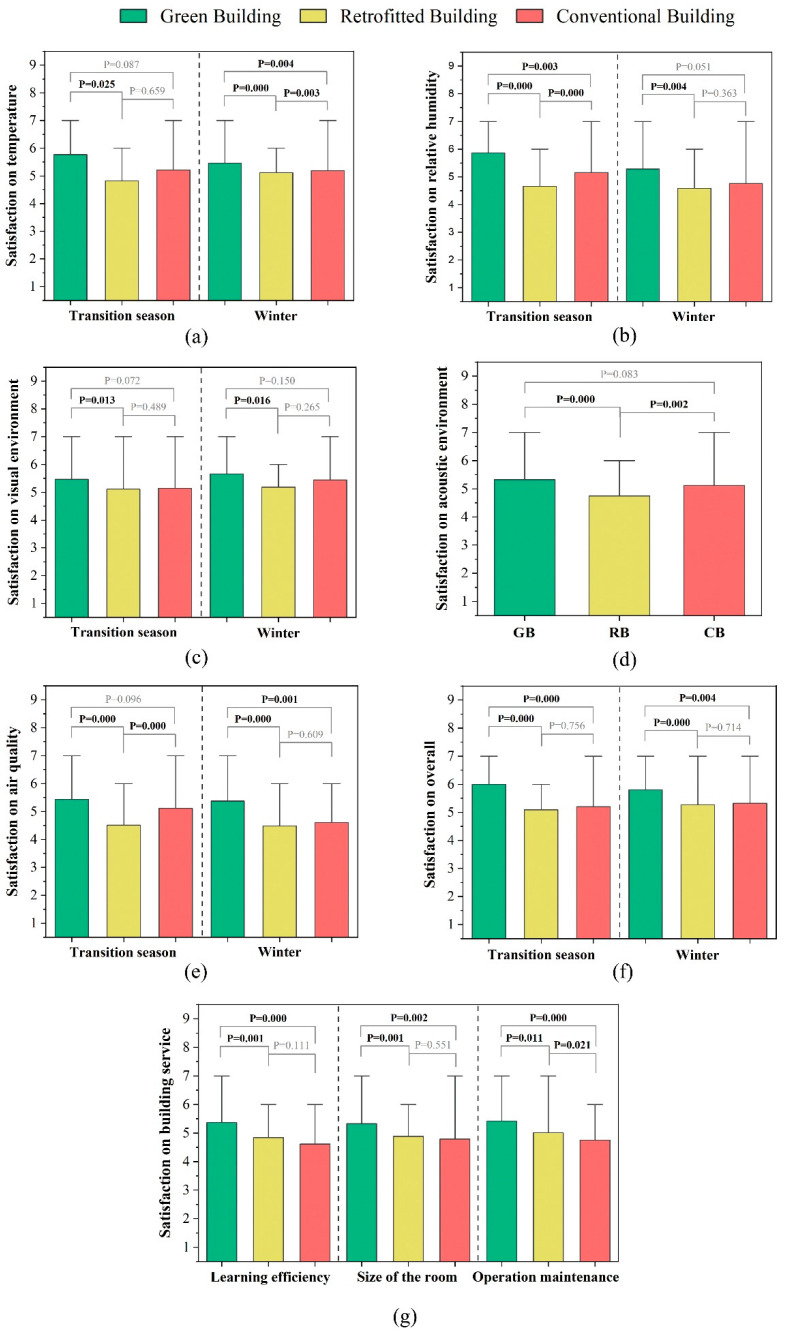
Comparison of occupant satisfaction with (**a**) temperature, (**b**) relative humidity, (**c**) visual environment, (**d**) acoustic environment, (**e**) air quality, (**f**) overall environment, and (**g**) building services in GB, RB, and CB.

**Figure 7 ijerph-20-00554-f007:**
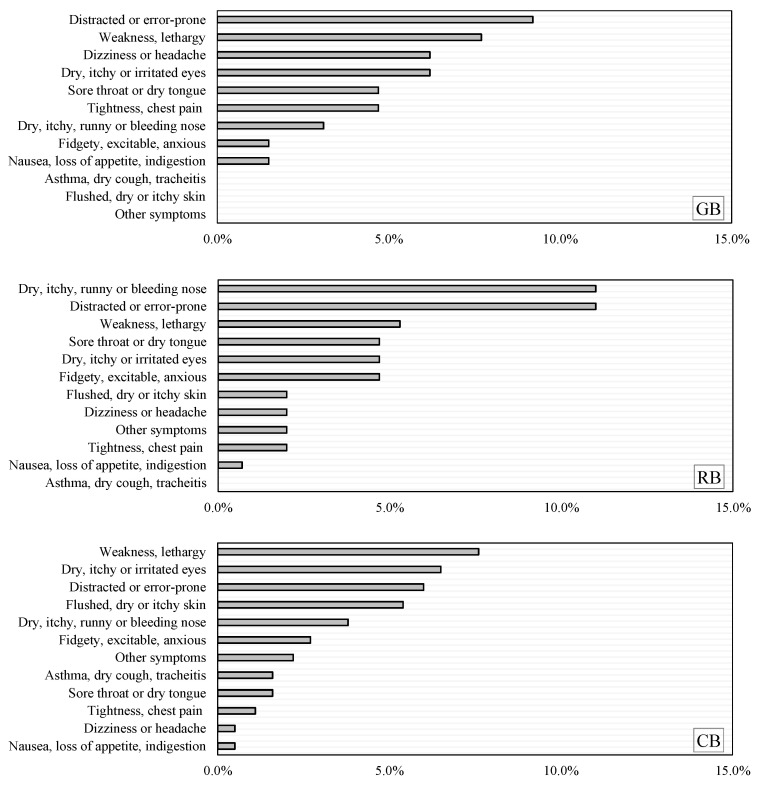
Comparison of the occurrence of SBS symptoms in the GB, RB, and CB.

**Table 1 ijerph-20-00554-t001:** Characteristics of the selected teaching buildings.

	Year Built or Retrofitted	Ventilation Mode	Gross Floor Area	Air Condition System	Number of Floors	Selected Classrooms
Green Building (GB)	2017	Mechanical	9297 m^2^	Central	5	12.6 × 7.2 m (South, 2nd floor) 7.2 × 8.4 m (East, 4th floor) 7.2 × 8.4 m (West, 3rd floor) 16.8 × 10.2 (North, 4th floor)
Retrofitted Building (RB)	2012	Mixed	3840 m^2^	Spilt	6	9.2 × 5.6 m (North, 1st floor) 9.2 × 5.6 m (South, 5th floor)
Conventional Building (CB)	1999	Natural	2904 m^2^	Spilt	6	10.0 × 6.5 m (North, 2nd floor) 10.0 × 6.5 m (North, 3rd floor) 10.0 × 6.5 m (North, 4th floor) 10.0 × 6.5 m (North, 5th floor)

**Table 2 ijerph-20-00554-t002:** Long-term and point-to-point indoor environment measurement approach.

	Thermal	Air Quality	Visual	Acoustic
Parameter	Air temperature and relative humidity	CO_2_ and PM_2.5_ concentration	Desktop illuminance	Sound pressure level
Location	One metre around the respondents		Desktop	One metre around the respondents at a height of 1.5 m
Long-term	Transition season, winter at least two weeks; 5-min interval		-	Continuous 20 min
Point-to-point	3-min average		The average of three measuring points	3-min average

**Table 3 ijerph-20-00554-t003:** Sensors and their accuracies.

Parameter	Sensor Model	Accuracy
Air temperature	SENSIRON SHT30-DIS	±0.1 °C
Relative humidity	SENSIRON SHT30-DIS	±1.5%
CO_2_ concentration	SenseAir S8 LP	±40 ppm
PM_2.5_ concentration	PMS70003	±10%
Illuminance	ROHM BH1750FVI	±4%
Sound pressure level	PR-ZS-BZ	±0.5 dB

**Table 5 ijerph-20-00554-t005:** Questionnaire content keywords.

	Content
Background information	Gender, age, and location
Satisfaction with an indoor environment	Temperature, humidity, visual, air quality and acoustics, learning efficiency, interior space design, and building service performance
Self-reported SBS symptoms	Dry, itchy, or irritated eyes Dry, itchy, runny, or bleeding nose Sore throat or dry tongue Tightness, chest pain Asthma, dry cough, tracheitis Flushed, dry, or itchy skin Weakness, lethargy Fidgety, excitable, anxious Dizziness or headache Nausea, loss of appetite, indigestion Distracted or error-prone Other symptoms

**Table 6 ijerph-20-00554-t006:** Compliance rates of different indoor environment parameters.

Parameter	GB		RB		CB	
Season	Transition	Winter	Transition	Winter	Transition	Winter
Air temperature	98.7%	20.6%	39.8%	79.8%	47.8%	36.7%
Relative humidity	96.6%	100%	84.0%	83.4%	84.7%	93.4%
CO_2_ concentration	78.5%	85.4%	100%	82.3%	100%	98.1%
PM_2.5_ concentration	65.3%	62.8%	44.6%	48.6%	48.6%	38.0%
Illuminance	100%	100%	40.4%	0%	21.6%	0%
Sound pressure level	79.6%		42.4%		58.7%	

**Table 7 ijerph-20-00554-t007:** Background information on respondents.

	Total	GB	RB	CB
Gender				
Male	163	53/82%	61/41%	49/23%
Female	236	12/19%	89/59%	135/77%
Age (years)				
<18	6	2/3%	0/0	4/2%
18–22	391	61/94%	150/100%	180/98%
>22	2	2/3%	0/0	0/0

**Table 8 ijerph-20-00554-t008:** Relationship between indoor environment and occurrence of SBS symptoms.

	Air Temperature	Relative Humidity	CO_2_ Concentration	PM_2.5_ Concentration	Illuminance	Sound Pressure Level
Dry, itchy, or irritated eyes	0.025	0.092	0.059	−0.045	−0.001	−0.044
Dry, itchy, runny, or bleeding nose	0.104 *	−0.111 *	−0.036	−0.021	0.046	0.000
Sore throat or dry tongue	0.123 *	−0.072	−0.066	−0.100 *	0.081	0.007
Tightness, chest pain	0.035	0.014	−0.039	−0.045	0.035	−0.015
Asthma, dry cough, tracheitis	−0.106 *	−0.069	−0.106 *	0.097	−0.071	0.017
Flushed, dry, or itchy skin	−0.106 *	−0.074	−0.118 *	0.099	−0.086	−0.002
Weakness, lethargy	0.085	0.042	0.057	−0.052	−0.017	0.027
Fidgety, excitable, anxious	0.015	−0.030	−0.061	−0.011	0.010	0.015
Dizziness or headache	0.028	−0.069	−0.011	−0.060	0.098	−0.020
Nausea, loss of appetite, indigestion	0.085	0.026	0.008	−0.086	0.004	0.017
Distracted or error-prone	−0.027	−0.015	−0.065	−0.087	0.055	−0.005
Other symptoms	0.015	0.014	0.010	−0.016	−0.062	−0.024

* Indicates significant correlation at the 0.05 level (two-tailed).

**Table 9 ijerph-20-00554-t009:** Relationship between indoor environment and learning efficiency.

	Air Temperature	Relative Humidity	CO_2_ Concentration	PM_2.5_ Concentration	Illuminance	Sound Pressure Level
Learning efficiency	0.006	−0.046	−0.128 *	0.16	0.148 *	−0.031

* Indicates significant correlation at the 0.05 level (two-tailed).

**Table 10 ijerph-20-00554-t010:** Relationship between the occurrence of SBS symptoms and learning efficiency.

	Learning Efficiency
Dry, itchy, or irritated eyes	−0.110 *
Dry, itchy, runny, or bleeding nose	−0.141 **
Sore throat or dry tongue	−0.105 *
Tightness, chest pain	0.026
Asthma, dry cough, tracheitis	−0.119 *
Flushed, dry, or itchy skin	−0.111 *
Weakness, lethargy	−0.169 **
Fidgety, excitable, anxious	−0.105 *
Dizziness or headache	0.006
Nausea, loss of appetite, indigestion	0.043
Distracted or error-prone	−0.087
Other symptoms	−0.095

** Indicates significant correlation at the 0.01 level (two-tailed); * Indicates significant correlation at the 0.05 level (two-tailed).

**Table 11 ijerph-20-00554-t011:** Results of multiple regression analysis.

Parameter	Unstandardised Coefficients		Standardised Coefficient	*t*	Sig.	95% Confidence Interval for B	
	B	Standard Error				Lower Limit	Upper Limit
(constant)	1.126	0.224		5.029	0.000	0.686	1.567
Temperature satisfaction	0.247	0.047	0.285	5.241	0.000	0.155	0.340
Humidity satisfaction	0.052	0.044	0.061	1.199	0.231	−0.033	0.138
Air quality satisfaction	0.103	0.044	0.110	2.341	0.020	0.016	0.189
Visual satisfaction	0.134	0.043	0.152	3.122	0.002	−0.050	0.218
Acoustic satisfaction	0.265	0.059	0.248	4.475	0.000	0.148	0.381

Note: The dependent variable is overall satisfaction.

**Table 12 ijerph-20-00554-t012:** Weightings of different indoor environment parameters.

Temperature	Humidity	Air Quality	Visual Environment	Acoustic Environment
*w* _T_	-	*w* _AQ_	*w* _V_	*w* _AC_
0.36	-	0.14	0.19	0.31

**Table 13 ijerph-20-00554-t013:** Comparison of satisfaction, IEQ, and occurrence of SBS symptoms.

	GB	RB	CB
Compliance rate	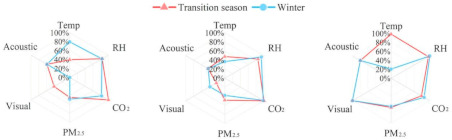		
*IEQ* _overall_	75.4	53.5	40.3
*IEQ* _T_	59.7	59.8	42.3
*IEQ* _AQ_	73	68.9	71.2
*IEQ* _V_	100	20.2	10.8
*IEQ* _AC_	79.6	42.4	58.7
*S* _overall_	5.6	4.8	5.1
Occurrence of SBS symptoms	26.2%	29.3%	24.5%

## Data Availability

The corresponding author will provide data from the paper when asked.

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
