# Peer review of "Field Measurements and Analysis of Indoor Environment, Occupant Satisfaction, and Sick Building Syndrome in University Buildings in Hot Summer and Cold Winter Regions in China"

_ijerph, 2022, doi:10.3390/ijerph20010554_

Round 1
Reviewer 1 Report
This study conducted a one-year post-occupancy evaluation study in three university teaching buildings to compare the performance differences in terms of the indoor environment, occupant satisfaction, SBS and other aspects of green, traditional, and renovated university buildings. These research results possess a certain reference value to the design and operating of the university teaching buildings. The structure of this paper is clear. The authors have carried out a relatively comprehensive and in-depth investigation and analysis, and found the performance differences between different buildings and the relationships.
The suggestions are as follows:
1. There are too many subsections in the section 3. It is recommended to merge them appropriately.
2. Add the introduction of local climate characteristics and the corresponding outdoor meteorological data. Add the comparison of Indoor/Outdoor value of PM2.5 concentration.
3. Add more latest papers to improve the quality of literature review, especially in the aspect of indoor environment of university building and the aspect of SBS.
4. Text and typographical errors. In table 15, the picture covers the table box. in table 3, the "acoustic" should be sound pressure level, and the accuracy should be ±0.5dB.
Author Response
Dear Professor and editor,
Thank you very much for your elaborate work with our paper. These valuable comments and suggestions have considerably improved the clarity of the paper. The manuscript has been revised in response to the comments with the main modifications listed below.
Reviewer #1:
General comments:
This study conducted a one-year post-occupancy evaluation study in three university teaching buildings to compare the performance differences in terms of the indoor environment, occupant satisfaction, SBS and other aspects of green, traditional, and renovated university buildings. These research results possess a certain reference value to the design and operating of the university teaching buildings. The structure of this paper is clear. The authors have carried out a relatively comprehensive and in-depth investigation and analysis, and found the performance differences between different buildings and the relationships. The suggestions are as follows:
- There are too many subsections in the section 3. It is recommended to merge them appropriately.
Reply: Thanks for your advice. We reorganized Section 3 and 4, combining several subchapters. We also deleted two tables to make the key results clearer.
- Add the introduction of local climate characteristics and the corresponding outdoor meteorological data. Add the comparison of Indoor/Outdoor value of PM5 concentration.
Reply: Thanks for your advice. We have added a figure to depicted the outdoor temperature change in Hangzhou in 2020. And a few sentences about the climate of the region were added. The average temperature in January is 7.4℃, the lowest is 0℃, the average temperature in July is 27.2℃, the highest is 37.0℃; July is sunny and hot with little rain and often dry. Annual precipitation is about 1500 mm. The comparison of indoor/outdoor(I/O) PM2.5 concentrations was added in the revised paper. It is found that I/O ratio in GB is much lower than CB and RB. I/O ratios in CB and RB fluctuate around one.
- Add more latest papers to improve the quality of literature review, especially in the aspect of indoor environment of university building and the aspect of SBS.
Reply: We have added several latest studies about the SBS in university buildings in line 112-126. SBS symptoms vary depending on comfort conditions such as hygiene, ventilation, and heating instead of the age of the school building. School principals responsible for the administration of school buildings have a marked impact in the improving or deteriorating of SBS symptoms [24]. Results in Iran showed that psychological factors such as job satisfaction, working environment, working hours and communication with col-leagues/employers were the most important factors affecting the prevalence of sick building syndrome [25]. Another study in Northern Iran suggested that there were significant correlations between CO2 and temperature with SBS symptoms [26]. Study from Eastern Mediterranean climate in educational laboratories indicated that SBS symptoms were associated significantly with education year and gender [27]. A higher CO2 concentration was significantly associated with a higher percentage of perceived stuffy odour and skin SBS symptoms in Chinese homes [28]. A study in Japan depicted that allergies and lifestyle behaviors were associated with increased SBS in children, including skipping breakfast, displaying faddiness, constipation, insufficient sleep, not feeling refreshed after sleep, and the lack of deep sleep [29].
- Text and typographical errors. In table 15, the picture covers the table box. in table 3, the "acoustic" should be sound pressure level, and the accuracy should be ±0.5dB.
Reply: Thanks for your advice. We have corrected these mistakes.
For any problem, please contact me, thank you.
Address: Zhejiang University City College, School of Spatial Planning and Design, Hangzhou, 310015, China;
Email: [email protected]

Reviewer 2 Report
The manuscript is well organized and well presenting what has been done. It presents an interesting topic that many readers around the globe would be interested in. Please find some comment below which would help you in revising the manuscript.
(1) Can you put some photos of the measurements took place at the three sites? It will help readers to understand the methods you conducted.
(2) Tables 8, 9, 10 show correlations that the indoor environments have with different responses of the participants. Given that one of your major research questions is to compare how different the results are in the three buildings; so, it would be better to present & compare the correlations between the buildings.
(3) Please use the values that your equations derived in the data analysis, and describe in the results.
(4) Discussion is a section for discussing the findings of the study. Please consider re-organise the tables & description about the results in this section (eg, Tables 13, 14, 15).
(5) Also, since there are quite a number of tables, it may be better to omit some tables or figures if they do not illustrate crucial findings.
(6) Please provide the information about the research ethics approval.
Author Response
Dear Professor and editor,
Thank you very much for your elaborate work with our paper. These valuable comments and suggestions have considerably improved the clarity of the paper. The manuscript has been revised in response to the comments with the main modifications listed below.
Reviewer #2:
The manuscript is well organized and well presenting what has been done. It presents an interesting topic that many readers around the globe would be interested in. Please find some comment below which would help you in revising the manuscript.
- Can you put some photos of the measurements took place at the three sites? It will help readers to understand the methods you conducted.
Reply: Thanks for your advice. Several photos of the measurements took place were added in the revised paper in Figure 1.
- Tables 8, 9, 10 show correlations that the indoor environments have with different responses of the participants. Given that one of your major research questions is to compare how different the results are in the three buildings; so, it would be better to present & compare the correlations between the buildings.
Reply: Thanks for your advice. We got the comparison results of SBS and work efficiency of different buildings before the submission, however, no relatively valuable results were obtained. Because there were too many sick building symptoms and too many influencing factors between different buildings. Therefore, we discussed the correlations between indoor environment, SBS and work efficiency in this section. In the future, we will consider expanding the sample sizes of buildings and questionnaire to discuss the differences of SBS performance between GB and CB in more detail later. The comparison of building performance in three buildings were summarized in the section 3.5 now.
- Please use the values that your equations derived in the data analysis, and describe in the results.
Reply: We have made changes in the results, and the values derived in the data analysis were updated.
- Discussion is a section for discussing the findings of the study. Please consider reorganise the tables & description about the results in this section (eg, Tables 13, 14, 15).
Reply: We have reorganized the table and description about the results in the discussion section. This part was moved into the results.
- Also, since there are quite a number of tables, it may be better to omit some tables or figures if they do not illustrate crucial findings.
Reply: According to your suggestion, we deleted table 11 and table 12 in section 3.
- Please provide the information about the research ethics approval.
Reply: More information about the research ethics approval was added in Institutional Review Board Statement.
For any problem, please contact me, thank you.
Address: Zhejiang University City College, School of Spatial Planning and Design, Hangzhou, 310015, China;
Email: [email protected]